# Methane Production in Soil Environments—Anaerobic Biogeochemistry and Microbial Life between Flooding and Desiccation

**DOI:** 10.3390/microorganisms8060881

**Published:** 2020-06-11

**Authors:** Ralf Conrad

**Affiliations:** Max-Planck-Institute for Terrestrial Microbiology, Karl-von-Frisch-Str. 10, 35043 Marburg, Germany; conrad@mpi-marburg.mpg.de

**Keywords:** methanogenesis, rice field soil, lake sediment, drainage, microbial community

## Abstract

Flooding and desiccation of soil environments mainly affect the availability of water and oxygen. While water is necessary for all life, oxygen is required for aerobic microorganisms. In the absence of O_2_, anaerobic processes such as CH_4_ production prevail. There is a substantial theoretical knowledge of the biogeochemistry and microbiology of processes in the absence of O_2_. Noteworthy are processes involved in the sequential degradation of organic matter coupled with the sequential reduction of electron acceptors, and, finally, the formation of CH_4_. These processes follow basic thermodynamic and kinetic principles, but also require the presence of microorganisms as catalysts. Meanwhile, there is a lot of empirical data that combines the observation of process function with the structure of microbial communities. While most of these observations confirmed existing theoretical knowledge, some resulted in new information. One important example was the observation that methanogens, which have been believed to be strictly anaerobic, can tolerate O_2_ to quite some extent and thus survive desiccation of flooded soil environments amazingly well. Another example is the strong indication of the importance of redox-active soil organic carbon compounds, which may affect the rates and pathways of CH_4_ production. It is noteworthy that drainage and aeration turns flooded soils, not generally, into sinks for atmospheric CH_4_, probably due to the peculiarities of the resident methanotrophic bacteria.

## 1. Theoretical Background

Environmental methanogenesis is the degradation of organic matter under anaerobic conditions to the gaseous products CH_4_ and CO_2_. The methanogenic pathway is catalyzed by a complex microbial community basically consisting of fermenting bacteria and methanogenic archaea. The entire process is anaerobic, since it only takes place in the absence of O_2_ and other oxidants (reducible inorganic electron acceptors). Therefore, it is not surprising that most reports of environmental CH_4_ production refer to anoxic environments, such as animal gut systems, aquatic sediments, flooded soils, peatlands and coastal wetlands [1,2,3,4,5,6,7,8]. More recently, however, it was found that the potential to produce CH_4_ under anaerobic conditions is widespread among various soil ecosystems, including non-flooded oxic upland soils [9]. Therefore, it is worthwhile to examine the consistency of our theoretical background with the occurrence of methanogenic processes.

Our understanding of environmental methanogenesis in anoxic environments is based on physicochemical and microbiological theories, of which the most relevant are the following. Chemically, complete degradation of organic matter to CO_2_ is usually an oxidation reaction, which primarily uses O_2_. The O_2_ required by the degradation process is supplied by diffusion from the atmosphere into the soil. The diffusion coefficient (D) in air is about four orders of magnitude larger (about 2 × 10^−1^ cm^2^ s^−1^) than that in soil water (about 2 × 10^−5^ cm^2^ s^−1^) [10]. Diffusion through x = 1 cm of water requires about t = 7 h, and through 10 cm of water even requires 70 h (calculated from t = x^2^/[2 D]). Therefore, it is plausible that oxidation processes in flooded soils and other wetlands become rapidly limited by O_2_. Once O_2_ is depleted, other oxidized inorganic compounds, such as nitrate, ferric iron, or sulfate can serve as oxidants. The propensity for using a particular oxidant depends on the redox potential of the electron-accepting reaction; i.e., degradation processes with the most negative Gibbs free energy will dominate [11,12]. The actual Gibbs free energy (ΔG) is the result of the ΔG° under standard conditions and the concentrations of the reactants and products involved. As a consequence, oxidation of organic matter usually results in the sequential reduction of available oxidants. In soils, the most common oxidants are O_2_, nitrate, ferric iron, and sulfate, which undergo sequential reduction along with the oxidative degradation of organic matter. Once inorganic oxidants are no longer available, however, anaerobic degradation of organic matter can only proceed by disproportionation resulting in the production of CO_2_ (oxidized compound) and CH_4_ (reduced compound). According to this theoretical picture, organic matter degradation to CH_4_ should not start before all the available inorganic electron acceptors have been depleted to such an extent that the ΔG of the oxidation reactions is less negative than the ΔG of the CH_4_-producing disproportionation reaction. Disproportionation of polymeric organic matter, e.g., cellulose, results in the formation of equal amounts of CO_2_ and CH_4_. Stoichiometry of methanogenic degradation of cellulose, or other organic compounds with a carbon oxidation state of zero, also constrains the pathways of CH_4_ formation resulting in >2/3 of CH_4_ being produced from acetate and <1/3 being produced from H_2_/CO_2_ [13,14,15].

The microbiological theory adds knowledge about the activity of individual microorganisms, which all together make up the environmental microbial community. Of course, the activity of microorganisms cannot overcome the laws of physics or chemistry, but the microbial enzyme machinery provides important catalysts without which most of the chemical processes would proceed very slowly or not at all. Therefore, microorganisms with particular catalytic functions must be present in sufficient numbers to allow a process to proceed. If such microorganisms are absent or inactive, processes may fail to proceed even when thermodynamic conditions are favorable. In soils with the potential for methanogenesis, important microbial functions include hydrolysis of complex organic matter, various fermentation processes, methanogenic processes and processes involved in the sequential reduction of inorganic compounds. Hydrolytic and fermenting bacteria degrade complex organic matter to H_2_, CO_2_ and simple organic compounds, which are subsequently further fermented to acetate, H_2_ and CO_2_. These compounds are the substrates for methanogenic archaea, which convert them to CH_4_ and CO_2_ [3,14,16,17]. Methyl compounds (e.g., methanol, trimethylamine, dimethylsulfide) are also possible substrates for methylotrophic methanogenesis, for example during degradation of pectin-producing methanol or in saline environments where methyl compounds are produced from osmolytes [14].

A sufficiently high water potential is a precondition for all microbial activity, which usually strongly decreases below −6 bar, although potential for growth of particular bacteria may be maintained down to −70 bar [18,19]. The decrease in water potential during the desiccation process can initiate the formation of microbial dormancy, including resting stages such as spores and cysts [20]. Dormancy will persist until the water potential becomes permissive again. Since many microorganisms, e.g., methanogenic archaea, do not form resting stages, it is unclear to which extent they can switch into dormancy and survive detrimental conditions. Microorganisms catalyzing the methanogenic degradation of organic matter are believed to be facultatively or obligately anaerobic, meaning that growth and/or activity of these microorganisms requires the absence of O_2_ and sometimes even a low (<−330 mV) redox potential [15]. Methanogenic archaea in particular are unable to proliferate in the presence of O_2_, and their activity is strongly inhibited [3].

Another important theoretical microbiological background concerns the concentrations of the growth substrates required for energy generation. Both the catalytic activities and the growth rates of microbial populations depend on substrate concentrations. The relationship is usually a hyperbolic one, described by the Michaelis–Menten and the Monod equations for catalytic activity and for population growth, respectively [21,22]. Both equations are parametrized with respect to their maximum rates (V_max_, µ_max_, respectively) and the substrate concentrations (K_m_, K_S_, respectively) at which the rates are half-maximal. The lower threshold (T) for utilization of the methanogenic substrate is another important parameter [23,24]. As a consequence, activity and growth of microorganisms depend on the concentrations of their substrates (both electron donors and acceptors), for which they may compete. The microorganisms with the better catalytic parameters (e.g., µ_max_, K_S_, T) will win the competition over a limiting substrate and, for example, outgrow a competitor. Please note that the result is comparable (but not identical) to that caused by chemical competition according to the ΔG values that are also dependent on the concentrations of substrates (but also on those of the products) according to Nernst law. For example, different hydrogenotrophic methanogenic species have different characteristic kinetic parameters, in particular thresholds for H_2_ consumption [25,26,27,28]. There are two groups of methanogens distinguished by their threshold concentrations for H_2_ that operate with different modifications of the CO_2_ reduction pathway [29]. There are also two groups of acetotrophic methanogens (*Methanothrix* spp. versus *Methanosarcina* spp.) having different enzyme systems for acetate activation and which consequently dominate at different acetate concentrations in the environment [30,31,32].

## 2. Flooding and Desiccation

Flooding and desiccation are key events for methanogenic processes in wetlands, which include peatlands [33], coastal wetlands [8] and flooded soils. In the following, the review will largely focus on flooded mineral soils, excluding peat soils and marine environments. Methanogenesis can only happen in the phase after flooding and before drainage. A standard model is rice field soils, which are seasonally submerged. Rice fields can undergo this cycle of flooding and desiccation once or twice, in some areas even three times a year [34]. However, there can also be multiple cropping, where cultivation of flooded rice is regularly rotated with cultivation of upland crops, such as maize, wheat or soya [35]. Hence, the time periods between upland conditions, which are more or less desiccated, and flooded conditions may vary between months and years. There are also natural areas, which undergo seasonal or irregular flooding, e.g., the flood plain of the Amazon River [36,37]. Moreover, there are lake sediments that are permanently flooded and, therefore, can represent a source for atmospheric CH_4_ [38]. On the other hand, there are so-called upland soils that are almost never flooded. At the dry extreme, there are desert soils with a low soil moisture content due to rare precipitation [39]. Theoretically, it is expected that methanogenic degradation of organic matter only operates under flooded conditions, when O_2_ and potential inorganic electron acceptors are all reduced. From a physicochemical point of view, this expectation is straightforward. However, from a microbiological point of view, it cannot be predicted that all the necessary functional types of microorganisms are actually present in all the different soil environments. This is particularly true for methanogenic microbial communities, which are expected to consist of anaerobic microorganisms that are sensitive to O_2_ and, thus, should not survive desiccation events.

After 1990, molecular methods for characterization of the composition of microbial communities and the abundance of microbial populations were quite well established and became more and more sophisticated and widespread. It has since then been possible to investigate both the function and the structure of the methanogenic microbial communities in environmental samples. Hence, it has been possible to test how the methanogenic process in various environments functions on the basis of both abiotic and microbial variables. This research resulted largely in the confirmation of our theoretical background knowledge, but also in a number of unexpected observations that required modification of our perspectives, in particular concerning the O_2_ sensitivity of microorganisms and the role of soil organic carbon (SOC) compounds. These aspects will be covered in the next three sections (Section 3, Section 4 and Section 5).

## 3. Observations in Accordance with Theory

The physicochemical and microbiological theories outlined in Section 1 are based on background knowledge and have repeatedly been tested and challenged by hypothesis-driven biogeochemical and physiological experiments and observations relevant for environmental methanogenic processes. The most important ones, in particular in the context of flooded rice field soils, are listed without extensive discussion or literature review in the following. (1) Flooding of soil initiates sequential reduction processes, which largely proceed in the sequence of reduction of nitrate, ferric iron, and sulfate before CH_4_ production starts [40,41]. (2) Vigorous CH_4_ production is suppressed as long as concentrations of the methanogenic substrates (H_2_, acetate) are kept low by the presence of reducible inorganic compounds [42]. (3) This suppression is consistent with thermodynamic conditions being unfavorable or favorable for CH_4_ production [43]. (4) Aeration of methanogenic soil results in regeneration of ferric iron and sulfate, thus suppressing CH_4_ production until these compounds are reduced again [44,45]. (5) Addition of nitrate, ferric iron or sulfate to methanogenic soils results in suppression of CH_4_ production, mainly due to creating adverse thermodynamic conditions [46,47]. (6) Addition of nitrate (like the addition of O_2_) results in the partial regeneration of ferric iron and sulfate, prolonging suppression of CH_4_ even when nitrate has been completely reduced [47]. (7) Organic matter (e.g., polysaccharides) is degraded sequentially by hydrolysis, fermentation, and hydrogenotrophic plus aceticlastic methanogenesis [48].

Similarly, many observations supported the microbiological theories, e.g., observations made in flooded rice field soils. (1) The sequential methanogenic degradation of organic matter is paralleled by changes in the composition of the microbial community initiated upon depletion of O_2_ [49,50,51]. (2) The absence or inhibition of functional microbial groups results in suppression of CH_4_ production. For instance, the inhibition of methanogens by bromoethane sulfonate or chloroform, or the specific inhibition of acetotrophic methanogens by methyl fluoride, results in accumulation of the respective precursor metabolites and allows mass balance calculations for fermentation processes or stable isotope fractionation [52,53,54]. (3) Production of CH_4_ in flooded soils requires the presence of a sufficient number of the microbes involved. Even more importantly, it is necessary that the microbes are not only present but express sufficient activity [55,56,57,58]. (4) The microorganisms involved in CH_4_ production are more or less sensitive to O_2_. This observation, in particular, requires closer inspection.

## 4. Oxygen Sensitivity and Microbial Populations in the Soil Environment

The most important microbiology hypothesis probably concerns the relation of microorganisms to O_2_, which is highly relevant for anaerobic microbial life. This hypothesis states that anaerobic microorganisms become active in anoxic flooded soil and inactive upon desiccation. Furthermore, obligate anaerobes might be killed after exposure to O_2_. The removal and exposure of soil to O_2_ is probably a more important consequence of flooding and drainage, respectively, than the water potential of the soil. Some obligately anaerobic microorganisms are also able to form resting stages, e.g., the endospore-forming *Firmicutes*, which survive by entering dormancy. Methanogenic archaea, however, do not form resting stages and, therefore, they were believed to be intolerant to desiccation and O_2_ exposure [15], albeit this demand was qualified early on [3,59]. Indeed, desiccation and/or aeration of anoxic soil results in cessation of CH_4_ production, albeit the numbers of methanogenic archaea usually do not decline completely [60]. In general, most soils were found to contain low, but significant populations of methanogenic archaea, which start to proliferate when soil is flooded and O_2_ and other inorganic electron acceptors have been depleted [61,62]. This picture has repeatedly been reproduced, demonstrating that even desert soils contain low populations of methanogenic archaea [39,63]. Additionally, pure cultures of methanogenic species (e.g., *Methanosarcina* sp.) were found to be not completely killed by exposure to O_2_, although CH_4_ production is inhibited [60]. Later on, it was found that although methanogenic archaea cannot grow in the presence of O_2_, they are able to survive and even partially utilize O_2_ as an electron acceptor [64,65]. Methanogenic archaea are apparently able to assimilate carbon even when soils are aerated [66,67]. However, not all methanogenic archaea are able to tolerate O_2_. The tolerant ones can be distinguished from the intolerant ones based on their functional gene content, and are hierarchically clustered into two classes, II and I, respectively [68]. Hence, the microbiology theory concerning O_2_ tolerance has to be modified, i.e., that some taxa of obligately anaerobic methanogens can survive (maybe even exploit to some extent) oxic conditions.

In the recent years, our group investigated the structure and function of methanogenic microbial communities in different soil environments that are distinguished according to the frequency of flooding. We hypothesized that these environments have characteristic populations of O_2_-sensitive microorganisms, in particular fermenting bacteria and methanogenic archaea. Thus, environments with more frequent drainage should contain more O_2_-tolerant and/or desiccation-tolerant microorganisms, such as class-II methanogens and endospore-forming *Firmicutes*. We also hypothesized that these populations and their biochemical functions change upon desiccation and reflooding in different ways according to the category of the soil environment. Thus, we expected that more frequently drained soil environments exhibit smaller population changes than permanently flooded environments upon drainage or permanently dry environments upon flooding. We distinguished four categories: (1) permanently flooded lake sediments [69,70]; (2) annually flooded rice field soils or river floodplains [71,72,73]; (3) irregularly flooded soils (rice crop rotation) [74,75,76]; and (4) upland soils (including desert soils) [39,62,72,73,74,77]. We checked the potential of CH_4_ production, the abundance of bacterial and archaeal microbiota, and the community composition of methanogenic archaea and (fermenting) bacteria. In the following, the observations will briefly be reviewed to demonstrate to what extent they are consistent with our hypotheses (Table 1).

Virtually all the different environments exhibited a potential for CH_4_ production provided the soil was submerged. Furthermore, all the studies demonstrated that potential CH_4_ production ceased upon drainage and desiccation of the soil. However, whereas CH_4_ production started immediately in permanently flooded lakes, the other soil environments exhibited a more or less pronounced lag phase. This lag phase was caused by preferential reduction of alternative electron acceptors (e.g., ferric iron, sulfate) until they were depleted, and by growth of anaerobic microorganisms (e.g., methanogenic archaea). Indeed, numbers of methanogenic archaea were typically low in dry upland soils or desert soil crusts, where they only increased upon flooding. By contrast, all the other soil environments always contained large populations of methanogens, which just required reducing conditions and sufficient substrate (H_2_, acetate) concentrations to become active (Table 1).

Once sufficiently large populations sizes of methanogens were established, drainage of soil did not result in immediate decrease, showing that the methanogens were relatively insensitive to desiccation and O_2_ exposure. Only prolonged drainage over several seasons resulted in a decrease in these putatively O_2_-sensitive populations [75,76,78]. The relative insensitivity of the methanogens to O_2_ is consistent with the observations that dry and occasionally flooded soil environments were dominated by methanogens belonging to the genera *Methanosarcina* and *Methanocella*, which both belong to the O_2_-tolerant class II [68]. *Methanosarcina* and *Methanocella* species also increased in relative abundance after desiccation of permanently or frequently flooded soil environments, which, however, also contained O_2_-sensitive methanogens of class I (e.g., *Methanosaetaceae*, *Methanobacteriales*) [71,73]. Hence, the abundance and composition of methanogen communities was consistent with theoretical expectations concerning O_2_ sensitivity. However, O_2_ sensitivity was apparently not the sole criterion. For example, in permanently flooded lake sediments [69], *Methanomicrobiales* belonging to class II decreased upon desiccation, and, in some paddy soils, [71,76] *Methanobacteriales* belonging to class I increased. Methanogens that were actively assimilating carbon in aerated soils belonged to methanogens of both class I and class II [66,67]. As a conclusion, O_2_ sensitivity is apparently not the sole explanation for the occurrence of methanogenic taxa; at least, it does not explain the population dynamics of *Methanosaetaceae*, *Methanomicrobiales* and *Methanobacteriales*, which seem to follow criteria other than O_2_ sensitivity.

The substrate supply of the methanogenic archaea is accomplished by fermenting bacteria. Endospore-forming *Firmicutes* were an important group of fermenting bacteria in virtually all anoxic soil environments (Table 1). It is not surprising that the relative abundance of *Firmicutes* was found to increase upon desiccation in most categories of flooded soil environments. *Firmicutes* was the major taxon being activated upon hydration of desert soil crusts [77] and increased in relative abundance after desiccation and reflooding of lake sediments [69] and of many other flooded soil environments [73]. Nevertheless, some environments contained *Firmicutes* as only a minor group of bacteria [76], and, in some environments, their relative abundance was not enhanced after desiccation [72]. Instead, *Acidobacteria*, *Actinobacteria*, *Chloroflexi* and *Proteobacteria* exhibited larger population dynamics. In summary, the abilities to be O_2_-tolerant and form resting stages are apparently important but not exclusive criteria for survival and activity of bacterial taxa in flooded and drained soil environments.

## 5. Role of Soil Organic Carbon

The CH_4_ production potential always recovered when drained soil was flooded again. However, once recovered, CH_4_ production rates were not necessarily equal to those before drainage or desiccation (Table 1). In some environmental systems, rates were decreased; in others, they increased. For example, CH_4_ production in upland soils and soils with crop rotation always decreased upon second flooding, whereas, in wetland rice fields, Amazonian wetlands and lake sediments, potentials increased, decreased or remained the same (Table 1). The reason for such changes cannot so far be traced to changes in the size or composition of the methanogenic microbial communities. Instead, it may be hypothesized that desiccation resulted in a change in the composition of the organic matter (e.g., by chemical reactions) and that such change affected the propensity of being degraded under anaerobic conditions [71,79]. In fact, the composition of SOC may be of similar importance for the functioning of the microbial methanogenic communities as their composition. The composition of SOC, for example, defines the recalcitrance of organic matter. Degradation of SOC to methanogenic substrates (e.g., H_2_, CO_2_, acetate) is arguably the rate-limiting step in the production of CH_4_ [80,81,82].

The apparent pathway of CH_4_ production in different soil environments was usually higher than 50% hydrogenotrophic methanogenesis, except in flooded rice fields, where it was close to the expected value of <33% (Table 1). The reason why the contribution of aceticlastic and hydrogenotrophic methanogenesis did not follow the expectation may be the activity of syntrophic acetate oxidation coupled to hydrogenotrophic methanogenesis, despite of the presence of populations of aceticlastic methanogens [83,84]. However, acetate can also be oxidized by reduction of organic compounds [85]. In fact, the soil organic matter may be degraded, involving redox reactions of SOC compounds [13,14]. It is known that SOC contains redox-active compounds, which may serve as electron donors, electron acceptors or electron shuttles between bacteria and iron minerals [86,87,88,89,90]. Literature data indicate that changes in the oxidation state of organic matter do happen [89,91,92,93,94,95], although resolution of individual carbon compound formulas does not presently allow identification of the compounds involved. Our knowledge of the composition and dynamics of the thousands of different environmental organic carbon compounds [95,96,97,98] is very limited and requires much more research in order for us to be able to amend our theoretical picture of methanogenic degradation pathways accordingly.

## 6. Methane Oxidation

Flooded soil environments usually have a strong potential for CH_4_ production and are generally a source of atmospheric CH_4_, whereas drained environments lose their potential for CH_4_ production and stop being a source (see Section 4). Literature indicates that many soil environments actually turn into a net sink of CH_4_ when drained, indicating that CH_4_-oxidizing microorganisms become active that are able to oxidize CH_4_ at low atmospheric concentrations. Such observations were made in soils from peat swamps [99,100], river bank margins [101] and acidic wetlands [102]. However, dried soils do not generally possess activity for aerobic CH_4_ oxidation, but require induction by incubation at sufficiently high CH_4_ concentrations [103]. Such soils also are not generally able to utilize the low concentrations of atmospheric CH_4_ [104]. Thus, drained rice fields usually do not become a net sink for atmospheric CH_4_ [105,106,107], and the CH_4_ oxidation potential ceases earlier upon drainage than the CH_4_ production potential [108]. Only flush feeding with sufficient amounts of CH_4_ was found to guarantee the development of specific methanotrophic bacteria able to oxidize atmospheric CH_4_ concentrations [109]. These high-affinity methanotrophs belonged to canonical genera of methanotrophic bacteria (e.g., *Methylocystis*, *Methylosarcina*). Desert soil has been found to become a net sink of atmospheric CH_4_ once water is sufficiently available, apparently because high-affinity methanotrophs are activated [110,111].

## 7. Conclusions

Flooding and drainage of soil environments control the availability of O_2_ and the operation of methanogenic microbial processes. Empirical studies of the structure and function of soil microbial communities confirm the validity of pertinent biogeochemical and microbiological theories, but also demonstrate some unexpected processes. In particular, it has become apparent that some groups of anaerobic microbes (e.g., methanogenic archaea) are not thus strictly anaerobic, as formerly believed. Some groups of methanogens were found to tolerate O_2_ to quite some extent and also to express this characteristic in various soil environments ranging from permanently flooded to dry upland soils. Without such substantial O_2_ tolerance, many wetlands could probably not act as a source for atmospheric CH_4_, so that natural and artificial wetlands would not constitute one of the most important sources (accounting for about 250–300 Tg a^−1^ or 50% in the global CH_4_ budget [112]. Biogeochemical and microbiological theories concerning environmental methanogenesis have to be amended accordingly. Other interesting observations concern the likely involvement of soil organic carbon compounds in redox processes and methanogenesis, so that for the functioning of environmental methanogenesis the composition of SOC may be of similar importance as that of the microbial community. Finally, it is interesting that drainage and aeration does not always turn a flooded soil environment into a net sink for atmospheric CH_4_. The reason for this behavior may be differences in the methanotrophic microbial communities among the various soil environments, which deserves further research.

## Figures and Tables

**Table 1 microorganisms-08-00881-t001:** Effects of desiccation or flooding on microbial populations and methanogenic functions in soils across a gradient from permanent wet to permanent dry conditions. (The symbols denote the following: = constant; ↑ increase; ↓ decrease).

	Permanent Wet	Seasonal Flooding	Seasonal Flooding	Rotation	Mostly Dry	Permanent Dry
	lake sediments	Amazon floodplain	paddy rice	rice—upland crop	upland soil	desert soil crusts
**Methanogen** numbers after flooding	=	=	=	=	↑	↑
**Methanogen** taxa stimulated by desiccation	*M’sarcinaceae* *M’cellaceae*	*M´sarcinaceae* *M´cellaceae*	*M´sarcinaceae* *M´cellaceae* *M´bacteriales*	*M´sarcinaceae* *M´bacteriales* or unchanged	*M´sarcinaceae* *M´cellaceae* *M´bacteriales*	*M´sarcinaceae* *M´cellaceae*
**Firmicutes** after desiccation and reflooding	↑	↑	↑ =	↑ =	↑	↑
**CH_4_ production** after desiccation and reflooding	↑	=	↑↓	↓	↓	
**Hydrogenotrophic** methanogenesis (%)	>50	>50	<33	<33	>50	>50
References	[69,70]	[73]	[71,72,76]	[74,75,76]	[39,72,73,74]	[62,77]

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
