# Peer review of "Methane Production in Soil Environments—Anaerobic Biogeochemistry and Microbial Life between Flooding and Desiccation"

_microorganisms, 2020, doi:10.3390/microorganisms8060881_

Round 1
Reviewer 1 Report
Flooding and desiccation are indeed the important factors influencing the methane production, which is sensitive to the oxygen level. This manuscript provides a sophisticated background on the theories of the biogeochemistry and microbiology regarding methanogenesis, summarizes literatures relevant to these theories, and emphasizes some new knowledge in terms of the oxygen tolerance of the methanogens, with the support of author’s works. This manuscript is well written and easy to understand. The literatures cited are up to date. It should be published after some revisions. I have only a few comments as below.
- Some important environments are not mentioned or discussed referring to flooding and desiccation. One such example is the massive drainage of peatlands and the followed restoration by rewetting, turning the ecosystem from methane sink to source. The other typical system subjected to flooding and dry cycle is the coastal tidal mangroves/wetlands, which is more complex but worthy being studied.
- The structure is a bit confusing. The first three parts talk about the well-known theoretical knowledge and summaries. From the fourth part, new information is talked, with relevance to oxygen toleration (part 4 and 5), and many more (part 6). I would suggest to re-organize these parts, especially P3 and P6, what are exactly the kind of observations? Are the other observations (P6) not in accordance with theory?
- The table provided is unclear and hard to read.
- Line 41-43. Please provide the reference.
- Line 59-62. What about the methylotrophic methanogenesis? They don’t account at all?
- Line 140-172. Are these observations in accordance with theory majorly in paddy soils? How about the other environments?
Author Response
1.I agree that flooded peatlands and coastal wetlands should also be mentioned. This is now done in the first paragraph (L.32-34) together with two additional quotations. However, the review will mainly focus on mineral soils and flooding by freshwater, which is now pointed out in L. 117-120
2.The structure is to first present the theoretical background for anaerobic methanogenic environments in general (section 1), then focus on soils flooded with freshwater (excluding peatlands and coastal wetlands)(section 2). Section 2 in the end emphasizes three points, which are covered in the following sections, i.e., (1) the confirmation of theory by observations (section 3), (2) the necessity to modify theory concerning O2 sensitivity based on observations (section 4), and (3) the necessity to extent theory according to the role of SOC (section 5). Finally, section 6 briefly addresses the role of CH4 oxidation. In the revision I try to explain this structure in a better way, also by revising the titles of the sections.
3.The table is now amended with an explanatory legend to make it better understandable.
4.A reference (Lerman 1979) is added. However, it is textbook knowledge, which can be found rapidly in the internet.
5.Methylotrophic methanogenesis is now briefly mentioned in L. 78-81
6.Please refer to point 2 for reply.
Reviewer 2 Report
The manuscript titled “Methane production in soil environments – anaerobic biogeochemistry and microbial life between flooding and desiccation” describes the previous and new insights on methane production, especially on oxygen-tolerant methanogens.
This is very useful in that it has been shown that methanogenic bacteria can survive in soils under field conditions.
General comments
Since the title says “ in soil environments”, it is important to describe the ecology of methanogen in soil, especially they are in microaggregates. Some studies indicated that methanogen can assimilate the carbon under the upland soil condition because they can act in anaerobic microaggregates (Lee et al. Appl. Soil Ecol. 2012; Watanabe et al. Appl. Soil Ecol. 2011).
L219-L227: Can you really say that methane production does not occur under drained conditions? Showing the true methane production under upland condition is difficult because methane is quickly consumed by methanotrophic bacteria. As mentioned above, some methanogens like Methanosarcina, Methanobacteria, and Methanocella can assimilate carbon under the upland condition and they may produce methane.
Section 6: I’m not sure what was the “other viewpoint”? The section is less informative compare with the previous section. What does this section mean?
Minor comments
L12 Noteworthy → Noteworthiness?
L64: What is BES?
L 145-159: some parts of this section were repeated of the previous section (L37-62). Do you need to separate them?
No comma after some adverbs or conjunctions.
Author Response
Thank you for the references. They are now added as arguments in section 4 in L. 202-204 and L. 254-255.
L.219-227. The observations to which the text refers generally showed that the potential for CH4 production ceased under drained conditions. This cannot be explained by CH4 oxidation, which is inactive under these conditions. Moreover, the lag phase of CH4 production caused by drainage is longer than the availability of O2. However, the point by the reviewer is taken up in the last section (section 6), where CH4 oxidation is discussed.
The titles of the sections , including “other viewpoints are now changed.
Minor comments
L.12. Noteworthy is the adjective of noteworthiness and is correctly used here.
L.164: BES is the abbreviation for bromoethane sulfonate, which is now spelled out in the text.
L.145-159: The previous section is about theory, the present one about observations supporting the theory; therefore, the separation.
I checked the text for commas, but (as non-native English speaker) may have made mistakes.